# CONTRASTIVE VISION TRANSFORMER FOR SELF-SUPERVISED OUT-OF-DISTRIBUTION DETECTION

## ABSTRACT

Out-of-distribution (OOD) detection is a type of technique that aims to detect abnormal samples that don't belong to the distribution of training data (or in-distribution (ID) data). The technique has been applied to various image classification tasks to identify abnormal image samples for which the abnormality is caused by semantic shift (from different classes) or covariate shift (from different domains). However, disentangling OOD samples caused by different shifts remains a challenge in image OOD detection. This paper proposes Contrastive Vision Transformer (CVT), an attention-based contrastive learning model, for self-supervised OOD detection in image classification tasks. Specifically, vision transformer architecture is integrated as a feature extracting module under a contrastive learning framework. An empirical ensemble module is developed to extract representative ensemble features, from which a balance can be achieved between semantic and covariate OOD samples. The proposed CVT model is tested in various self-supervised OOD detection tasks, and our approach outperforms state-of-the-art methods by 5.12% AUROC on CIFAR-10 (ID) vs. CIFAR-100 (OOD), and by 9.77% AUROC on CIFAR-100 (ID) vs. CIFAR-10 (OOD).

## 1 INTRODUCTION

As many deep neural networks (DNNs) are deployed in real-world applications, the safety and robustness of the models get more and more attention. Most existing DNNs are trained under the closed-world assumption, i.e., the test data is assumed to be drawn i.i.d. from the same distribution as the training data (Yang et al., 2021). Although the deployed DNNs can perfectly deal with such ID samples, they would blindly classify the data coming from other classes or domains (i.e., OOD samples) into existing classes in an open-world scenario. Nguyen et al. discovered that neural networks can be easily fooled by unrecognizable images, which means that most DNNs are unreliable when encountering unknown or unseen samples. Such a few mistakes may be tolerable in some scenarios (e.g., chatbot, interactive entertainment), whereas they will bring catastrophic damage when the application area requires great safety benefits, such as automated vehicles, medical imaging and biometric security system. Therefore, it is essential to equip the model with the ability of detecting out-of-distribution data and make it more robust and reliable.

Generally, the outlier arises because of the mechanical failure, fraudulent behaviour, human error, instrument error and natural deviations in populations (Hodge & Austin, 2004). In the field of machine learning, compared with ID samples, OOD samples are regarded as the outliers due to distributional shifts. The distributional shifts can be caused by semantic shift (i.e., OOD samples from different classes) or covariate shift (i.e., OOD samples from different domains) (Yang et al., 2021). Meanwhile, the OOD samples that are semantically and stylistically very different from ID samples are referred to as far-OOD samples, and those that are semantically similar to ID samples but different from ID samples in domains are referred to as near-OOD samples (Ren et al., 2021). The out-of-distribution detection, also known as outlier detection or novelty detection, is developed to identify whether a new input belongs to the same distribution as the training data. A natural idea is to build a classifier to identify the ID and OOD data, using such as Deep Neural Network (DNN) and Support Vector Machine (SVM). However, the sample space of OOD data is almost infinite as OOD dataset is the complementary set of ID dataset, which leads to that creating a representative OOD dataset is impracticable. Moreover, OOD samples are scarce and costly in some industries (e.g., medical imaging, fraud prevention). These are main issues in the research on OOD detection.

To address these problems, researchers focus on the latent features of ID data, assuming distinguishable distributional shifts exist between ID and OOD samples in the latent feature space. Some researchers (Nalisnick et al., 2019; Serrà et al., 2019; Xiao et al., 2020) use generative models, like Variational Auto-encoders (VAE), to extract the latent features for both ID and OOD samples, and specific OOD socres are designed and used as the metric. As an alternative, contrastive learning models can be employed to learn the latent features, such as Self-Supervised Outlier Detection (SSD) (Sehwag et al., 2020) and Contrasting Shifted Instances (CSI) (Tack et al., 2020). However, in contrastive learning, researchers usually adopt standard convolutional neural network (CNN) and its variants like ResNet (He et al., 2016) as the encoder. By contrast, the transformer-based architectures (such as the earliest Vision Transformer (ViT) (Dosovitskiy et al., 2020), DeiT (Touvron et al., 2021) and Swin Transformer (Liu et al., 2021)) gradually outperform CNNs in terms of extracting robust latent features as they can learn global long-range relationships for visual representation learning, which would facilitate the identification of ID and OOD samples.

In this paper, a Contrastive Vision Transformer (CVT) model is proposed for OOD detection under self-supervised regime for image classification tasks. The framework of contrastive learning, including data augmentation and contrastive loss, is adopted to learn the representation for all inputs, which has been shown to be reasonably effective for detecting OOD samples (Tack et al., 2020). On this basis, four extra modules are introduced into this framework: (i) To improve the distinguishability between ID and OOD samples in the latent space, vision transformer architecture rather than CNN is embedded as a feature extracting module; (ii) Since the collapse of representation is a noteworthy problem in self-supervised and unsupervised scenarios, an additional predictor structure (inspired by BYOL (Grill et al., 2020)) is employed to avoid collapsed solutions. (iii) Considering that the size of negative samples plays an important role in contrastive learning, a memory queue scheme from MoCo (He et al., 2020) is integrated to maintain the model's performance especially when the batch size is extremely small. (iv) An ensemble module is developed to build representative ensemble features for achieving the balance between semantic and covariate OOD detection, as we observe that in our experiments the latent features from the encoder perform better on semantic OOD samples but on the contrary the latent features from the predictor perform better on covariate OOD samples. To further improve performance, a Mahalanobis distance-based OOD score function is utilised for the OOD detection, the effectiveness of which has been shown in recent papers (Sehwag et al., 2020; Ren et al., 2021).

To conclude, the key contributions of the paper are as follows:

- We integrate vision transformer architecture into a contrastive learning framework and develop a new paradigm specifically for self-supervised OOD detection in image classification tasks, results outperform state-of-the-art algorithms
- We develop an ensemble module to compute representative features that balance OOD samples from different types of data shifts
- We conduct extensive ablation studies to report the influences of various hyper-parameters on OOD detection tasks and benchmark the performance of CVT using different vision transformer modules including ViT, ResNet50, and Swin transformer

In the rest of the paper, related work is described in Section 2 and the main CVT model is introduced in Section 3. Followed by numerical results in Section 4 and the paper is concluded in Section 5.

## 2 RELATED WORK

Contrastive learning is a self-supervised technique that has seen fast development in recent years. Chen et al. (2020) proposed a contrastive learning framework consists of four components: data augmentation module, neural network base encoder, MLP (multilayer perceptron) projection head, and contrastive loss. It incorporated a strong inductive bias by gathering samples from the same class and repelling others and achieved promising results in visual representation learning. Under a similar paradigm, many influential variants were develooped in recent years, such as SimCLR (Chen et al., 2020), MoCo, SwAV (Caron et al., 2020), BYOL and MoCo-v3 (Chen et al., 2021). MoCo introduced a queue module to store the key representations of negative samples since the number of negative samples can effectively improve performance. To maintain the consistency of keys in the queue, a momentum strategy was developed for MoCo to update the parameters of the key encoder.

BYOL adopted an asymmetric structure to prevent the model from collapsing by adding a extra predictor module after the projector for the online networks.

Current practice in contrastive learning adopts standard convolutional neural networks (CNN) like ResNet(He et al., 2016) as the encoder for image feature extraction. However, the attention-based architectures has recently outperformed CNN architectures in major vision tasks such as image classification, object detection, and semantic segmentation. It has been shown that attention module is capable of extracting global long-range relationships for visual representation learning. For example, ViT(Dosovitskiy et al., 2020) applied standard Transformer encoder to images classification tasks and reported good performance. It chopped a image into sequences of patches to fit the sequence-based Transformer model. It also demonstrated that the multi-head self-attention (MSA) was beneficial for representation learning in images classification. To achieve excellent results, ViT required large training data and extensive computing resources. DeiT(Touvron et al., 2021) introduced a distillation procedure (a teacher-student strategy) to enable Transformer learning from a convnet and shrinking the training time. To bring the benefit of self-attention to object detection and semantic segmentation tasks, Swin Transformer(Liu et al., 2021) employed a hierarchical architecture to produce features at various scales. It is worth to mention that, the shifted windowing approach in Swin Transformer brought a linear computational complexity compared to standard self-attention.

The generalised OOD detection operations were proposed by Yang et al. (2021) where they divided the methodology into four categories: classification-based, density-based, distance-based and reconstruction-based methods. The classification-based method usually applies a softmax function to the output layer, which enables the model producing a probabilistic result for all classes. Hendrycks & Gimpel (2016) observed that a well-trained model can give higher probability to ID samples than OOD ones. Based on the findings, ODIN (Liang et al., 2017) adopted temperature scaling in softmax function to separate ID/OOD softmax scores under standard classification models. However, the model may sometimes assign a high probability on known classes for OOD samples. To address this issue, Dirichlet-based uncertainty (DBU) (Kopetzki et al., 2021) incorporated uncertainty estimates by predicting the parameters of a Dirichlet distribution. An outstanding advantage of DBU models is that it can effectively compute epistemic distribution, aleatoric distribution, and class labels. With respect to density-based method, it established probability distribution of ID examples and placed the OOD samples in low-density areas. Pidhorskyi et al. (2018) utilised autoencoder network to capture underlying structure of data distribution and computed novelty probability. Compared to classification-based approaches, generative models usually had a worse performance (Yang et al., 2021). The idea of distance-based methods was that the distance from OOD input to ID samples may be relatively farther than distance from ID input to ID samples. People can leverage distances with Euclidean distance, geodesic distance, cosine similarity, or Mahalanobis distance between the feature embeddings of input and the centroids of all classes. SSD (Sehwag et al., 2020) showed that the Mahalanobis distance is effective in OOD detection based on a contrastive learning framework. Another OOD detector, CSI Tack et al. (2020) was also based on contrastive learning and it used distribution-shifting augmentations (e.g., rotations) to promote OOD detection.

## 3 METHODOLOGY

This section details the architecture of the proposed CVT model, a simple yet efficient self-supervised OOD detector, followed by the design of loss functions and evaluation metrics.

### 3.1 MODEL ARCHITECTURE

In some previous research (Tack et al., 2020; Ren et al., 2021; Yang et al., 2021), OOD detection is divided into two stages: first, a representation learner is built and trained to extract latent features; second, an OOD score (e.g., softmax probability and distance-based score) is computed as the metric to indicate whether a sample is an OOD one. As shown in Fig. 1, the general architecture of our CVT model is composed by two parts: (i) a contrastive representation learner based on contrastive learning framework, and (ii) a Mahalanobis distance-based OOD score function.

The contrastive representation learner consists of two parallel networks: an *online network*, whose parameters are updated by contrastive loss and back-propagation, and a *target network*, where a

moving average of the corresponding online parameters is used as its own parameters (Grill et al., 2020). Unlike the traditional contrastive learning model, all the encoder modules in our CVT are flexible and can be replaced by any DNN designed for extracting latent features. Here, the ViT (Fig. 2) is given as an example for the encoder module, and a pre-trained version is preferrd since the training of it requires a lot of samples. Furthermore, an additional predictor structure in BYOL is employed in the online network to avoid collapsed solutions, as well as a memory queue scheme in MoCo is integrated into the target network to maintain the model's performance under different batch size settings. Although MoCo-v3 abandoned the memory queue design and used a large batch size to train the model, we still keep it available especially on low-memory computing infrastructure.

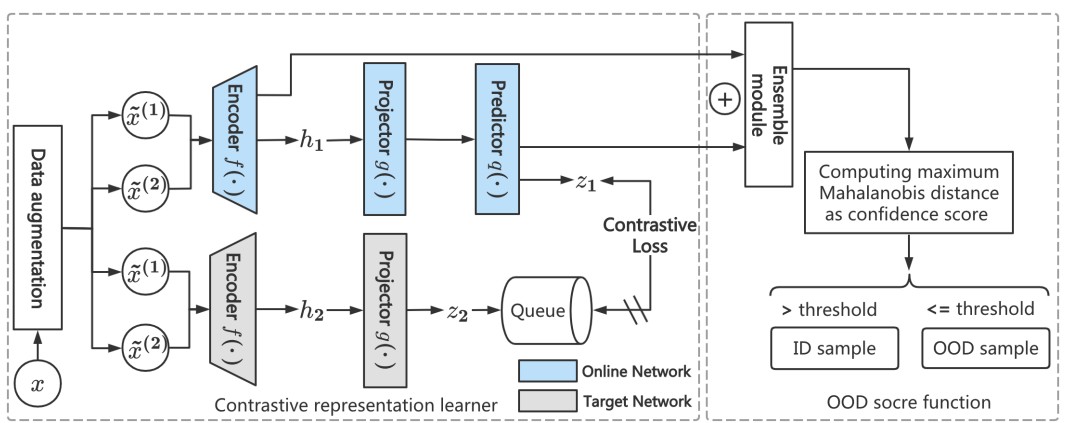

Figure 1: Architecture of the CVT model, in which the left part is a contrastive representation learner composed by two parallel deep networks (online and target networks highlighted in blue and grey respectively) with attention-based encoder $f(\cdot)$, projector $g(\cdot)$, and/or predictor $q(\cdot)$. The ensemble module computes output features from Encoder ($h_1$) and Predictor ($z_1$), then send ensemble features for OOD score computation on the right of the figure.

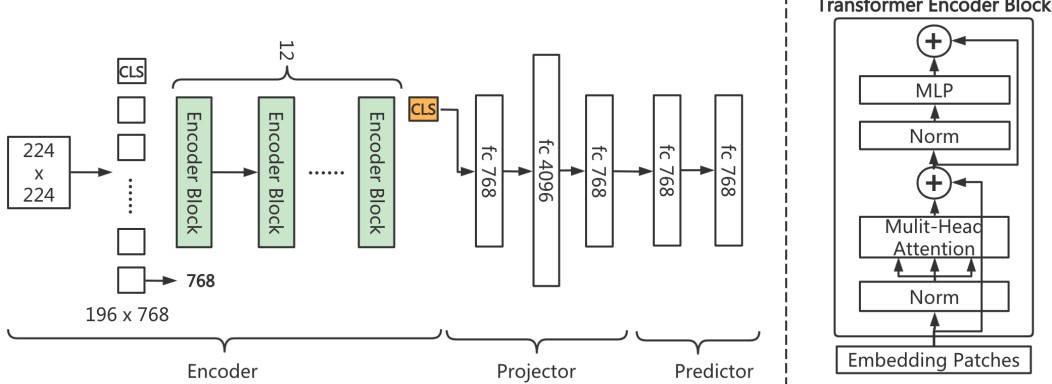

Figure 2: Design of the contrastive representation learner in Fig. 1. CLS is a placeholder to be used for prediction in downstream tasks and fc represents fully connected layer. Numbers in the figure indicate the dimension of the corresponding input/output. The encoder block highlighted in green is detailed on the right panel of the figure. Here we take a standard Transformer encoder as an example, however, the encoder block is flexible for other (non) attention-based architectures.

Given a dataset $\{x_i\}_{i=1}^N$ without labels, two views $\{\tilde{x}_i^{(1)}\}_{i=1}^N$ and $\{\tilde{x}_i^{(2)}\}_{i=1}^N$ of each sample are generated with data augmentation by using random ensemble transformations (Grill et al., 2020), and $2N$ samples are obtained in total. Then, the augmented data will be fed into the model in batches. For each view pair of the sample, $(\tilde{x}^{(1)}, \tilde{x}^{(2)})$, the representation pairs of $(z_1^{(1)}, z_1^{(2)})$ and $(z_2^{(1)}, z_2^{(2)})$ are produced by the predictor $q(\cdot)$ in the online network and the projector $g(\cdot)$ in the target network respectively. Meanwhile, two memory queues, $Q_1$ and $Q_2$, are created in the target

network to store the presentations $z_2^{(1)}$ and $z_2^{(2)}$ respectively. The contrastive loss is calculated among the representations given by the two networks (Section 3.2). While predicting, only the online network is used to generate the representations for each sample (no data augmentation).

For the OOD score function, we use the ensemble features to compute the maximum Mahalanobis distance between a new input and the clusters of the training set in latent space. The high-level features (e.g., from the predictor) usually contain more semantic information and low-level features (e.g., from the encoder) have more general properties. The semantic information helps distinguish near-OOD samples but fails on far-OOD samples. General properties can contribute to the identification of samples from different domains. Therefore, an ensemble module in the online network, where the element-wise mean of features/representations from the encoder and predictor is computed, is developed to build representative ensemble features for achieving the balance between semantic and covariate OOD detection. To keep the ensemble module feasible, we set the dimension of the output from the predictor to be the same as that from the encoder.

## 3.2 Loss Function Design

For unlabeled training data, the representations can be extracted by contrastive learning models. Our model absorbs the strengths of MoCo, MoCo-v3, and BYOL, and employs InfoNCE (Oord et al., 2018) as our training objective. The total loss is described as follows:

$$\mathcal{L}_{total} = \mathcal{L}_{z_1^{(1)}} + \mathcal{L}_{z_1^{(2)}} \tag{1}$$

$$\mathcal{L}_{z_1^{(1)}} = -\log \frac{\exp(z_1^{(1)} \cdot z_2^{(2)}/\tau)}{\sum_{i=0}^{K} \exp(z_1^{(1)} \cdot z_2^i/\tau))} \tag{2}$$

$$\mathcal{L}_{z_1^{(2)}} = -\log \frac{\exp(z_1^{(2)} \cdot z_2^{(1)}/\tau)}{\sum_{i=0}^{K} \exp(z_1^{(2)} \cdot z_2^i/\tau))} \tag{3}$$

where $z_2^i$ is the embedding stored in $queue_1$ and $queue_2$, $K$ is the sum of the size of both queues and $\tau$ is hyper-parameter temperature.

After the training, we just use the online network to extract embeddings from new inputs. Given another two dataset $\{x_i^{id}\}_{i=1}^{M}$ and $\{x_i^{ood}\}_{i=1}^{M}$, our OOD score function should assign a value to each input. we choose distance-based method to compute the scores, which is the Mahalanobis distance from input to centroids in training dataset. Therefore we have to divide training data into $C$ clusters. Here we use k-means clustering approach to classify every training samples to a particular class. Finally, we consider the maximum Mahalanobis distance as the OOD score, show in Eq. (4).

$$\text{score}(x) = \max_{c \in C} -(x - \mu_c)\hat{\Sigma}_c^{-1}(x - \mu_c)^T \tag{4}$$

where $c$ is the class of training data, $x$ is the embedding of input samples, $\mu_c$ is the mean of training data from class $c$.

## 3.3 Evaluation Metrics

As shown in Fig. 1, we define an OOD score based threshold to classify whether a new sample belongs to OOD or not. Specifically, a sample is regarded as an OOD sample if its OOD score is larger than the predefined threshold, otherwise it is treated as an ID sample.

We adopt three commonly used metrics to evaluate the performanc eof OOD detector. AUROC, which is a threshold-independent evaluation method and can reflect the ability of model to discriminate between positive examples and negative examples. A high AUROC value means the model has good discriminatory ability and an AUROC of 0.5 corresponds to a random guess (useless model). Area Under the Precision-Recall (AUPR) is another evaluation metric that represents the ability to correctly distinguish positive samples without treating positives as negatives. The third one, FPR95 represents the false positive rate (FPR) when the true positive rate (TPR) equals to $95\%$. It measures the size of OOD samples that are not correctly recognised when 95% of ID samples are correctly classified.

## 4 EXPERIMENTS

We follow commonly used benchmarks from previous work and consider CIFAR-10 and CIFAR-100 (Krizhevsky et al., 2009) as ID datasets. For the OOD dataset, we choose Describable Textures Dataset (DTD) (Cimpoi et al., 2014) and SVHN (Netzer et al., 2011) in order to compare the CVT model with other competing OOD detectors. ID samples are divided into ID training set and ID test set, with data leakage issue considered. For all experiments, models are trained with the ID training set in the training phase, and tested with both ID test set and OOD set in the evaluation phase. All the experiments are repeated 5 times by using different random seeds. We evaluate our model by three metrics: AUROC, AUPR, and FPR95.

In our benchmarking and ablation studies, we benchmark the performance of different representative base encoder networks under the CVT model, including ViT-B-16 (Dosovitskiy et al., 2020), a pre-trained ViT variant on Imagenet2012 (Russakovsky et al., 2015); Swin Transformer, the popular and powerful transformer architecture; and ResNet50, the basic but widely used CNN architecture. We train the proposed CVT model using optimizer AdamW(Loshchilov & Hutter, 2018) with an initial learning rate 1e-6, half-cycle cosine decay, weight decay 0.1, and batch size equals to 64. Following the setup in literature SSD (Sehwag et al., 2020), we set the default temperature to 0.5 and ablate different temperature and number of clusters pairs, see Section 4.5 for more details. All experiments are run on Ubuntu 12.04 with two NVIDIA RTX A4000 GPU cards. We use Python 3.8.10 and the single queue size in CVT is set to 4096.

### 4.1 CVT MODEL PERFORMANCE

As shown in the first row of Table 1, we test 6 pairs of combinations of ID and OOD datasets. Both CIFAR-10 and CIFAR-100 datasets shares common image characteristics (e.g., format, quality, style etc.), while SVHN dataset and Describable Texture dataset are more dissimilar to them.

We first compare CVT performance in self-supervised OOD detection tasks, where models are trained with unlabeled training data. We compare AUROC performance (higher score is better) across 6 different models including PixelCNN++(Salimans et al., 2017), Deep-SVDD(Ruff et al., 2018), Rotation-loss(Komodakis & Gidaris, 2018), CSI(Tack et al., 2020), and SSD(Sehwag et al., 2020) (shown in the top part of Table 1). The results demonstrates that the proposed CVT model clearly outperforms all other competing models. The averaged AUROC of CVT model has improved by 6.6% when comparing with the the 2nd best one (i.e., SSD). In the challenging scenario where CIFAR-100 is the ID set and CIFAR-10 serves as the OOD set, most existing OOD detectors perform poorly due to the similarity and size of CIFAR-100 over CIFAR-10. Our CVT model achieves a noteable score of 79.37%, which is almost 10% improvement from that of the 2nd best model.

Table 1: Model comparison in AUROC with 6 **self-supervised** OOD detectors using **unlabelled** training data and 5 **supervised** OOD detectors using **labelled** training data. The best performed number is highlighted in bold for each ID/OOD pair. The last column reports the averaged AUROC over all available tests (when the number of tests is no less than 4).

| Method | ID ACC (on CIFAR-10) | ID: CIFAR-10 vs. OOD: | | | ID: CIFAR-100 vs. OOD: | | | Average |
| --- | --- | --- | --- | --- | --- | --- | --- | --- |
| | | SVHN | CIFAR-100 | Texture | SVHN | CIFAR-10 | Texture | |
| PixelCNN++ | - | 15.8% | 52.4% | - | - | - | - | - |
| Deep-SVDD | - | 14.5% | 52.1% | - | 16.3% | 51.4% | - | 33.6% |
| Rotation-loss | - | 97.9% | 81.2% | - | 94.4% | 50.1% | - | 80.9% |
| CSI | 94.38% | 99.8% | 89.2% | - | - | - | - | - |
| SSD | 95.07% | 99.6% | 90.6% | 97.6% | 94.9% | 69.6% | 82.9% | 89.2% |
| CVT (ours) | $97.92_{\pm0.15}$% | $99.90_{\pm0.004}$% | $96.72_{\pm0.001}$% | $100_{\pm0}$%* | $98.78_{\pm0.001}$% | $79.37_{\pm0.002}$% | $100_{\pm0}$%* | **95.80%** |
| CE+SimCLR[†] | - | 99.5% | 92.9% | - | 95.6% | 78.3% | - | 91.6% |
| CSI[†] | $96.1_{\pm0.1}$% | $97.9_{\pm0.1}$% | $92.2_{\pm0.1}$% | - | - | 70.74% | - | - |
| SSD+[†] | 94.55% | **99.9%** | 93.4% | 98.5 | 98.2% | 78.3% | 81.2% | 91.6% |
| CIDER[†] | 94.63% | 99.73% | 93.04% | 97.44% | 97.75% | 77.02% | 91.96% | 92.82% |
| Fort et al.[†] | 98.70% | - | **98.52%** | - | - | **96.23%** | - | - |

* The 5 repeated results using different random seeds are all 100%. [†] The supervised OOD detection method that uses ID labels for training.

We also compares CVT model against supervised OOD detectors, where models are trained with labelled ID training set and tested by both unseen ID test set and OOD set. This is an unfair comparison to the CVT model as it is still trained by unlabelled (but the same) ID training set, i.e., no label information has been provided to CVT model. Surprisingly CVT still outperforms majority of the supervised OOD detectors except model in (Fort et al., 2021). As shown in the bottom part

of Table 1, we benchmark CE+SimCLR(Winkens et al., 2020), supervised CSI, supervised SSD, CIDER(Ming et al., 2022), and (Fort et al., 2021) in the two pairs of CIFAR datasets. To briefly conclude, the proposed CVT model performs well in different OOD detection tasks for both near/far OOD datasets.

## 4.2 COMPARISON BETWEEN ENSEMBLE FEATURES AND OTHER FEATURES

As depicted in Fig. 1, we develop an ensemble module to balance the features extracted from both encoder and predictor. In this subsection, we test and show the effectiveness of the ensemble module in the CVT model.

Table 2: Performance of ensemble module in CVT. 'Encoder features' and 'Predictor features' represent output features produced by the Encoder component and the Predictor component in the CVT model, respectively. 'Ensemble features' describes the merged features computed on both encoder/predictor features.

| ID dataset | OOD dataset | Encoder features | Predictor features | Ensemble features |
|---|---|---|---|---|
| CIFAR-10 | CIFAR-100 | 94.89% | **96.56**% | 96.06% |
| | SVHN | **99.95**% | 99.62% | 99.93% |
| CIFAR-100 | CIFAR-10 | 78.00% | 77.26% | **80.34**% |
| | SVHN | 99.06% | 95.41% | **99.21**% |

The first two columns in Table 2 show four test settings that are grouped into semantic shift scenarios (e.g., CIFAR-10/CIFAR-100 vs. SVHN) and covariate shift scenarios (e.g., CIFAR-10 vs. CIFAR-100). By using the features from either the Encoder or the Predictor to compute the OOD score for detecting OOD samples, the results display different trends under the above two scenarios. The race of the OOD detection under the covariate shift scenario between low-level features (i.e., the features from the encoder) and high-level features (i.e., the features from the predictor) ends in a draw. By contrast, low-level features dominate the OOD detection under the semantic shift scenario. Therefore, it is hard to decide which kind of features should be finally used to compute the OOD score. However, by comparing the performance of ensemble features with others under CIFAR-10 ID dataset, one sees that the ensemble module balances the performance between low-level and high-level features. Interestingly, one also observes that ensemble features achieves better performance under CIFAR-100 ID dataset. We reckon the ensemble features not only balance features but also enhance features for OOD detection.

## 4.3 ENCODER USING VIT VS. RESNET VS. SWIN TRANSFOMER

To evaluate our hypothesis that the attention-based models are capable of extracting more effective features for OOD detection, we compare the performance between three architecture options for the encoder module in CVT, namely ResNet50, ViT-B-16 (Base model of ViT), and Swin-B (Base model of Swin Transfomer). In this experiment, we calculate OOD scores using features output from predictor and the number of dimensions is set as 768.

Fig. 3 shows experimental results of using CIFAR-10 as the ID set while CIFAR-100 (left panel) and SVHN (right panel) as the OOD set. One sees that both ViT and Swin Transformer architectures clearly outperform standard ResNet50 by about 26% in both AUROC and AUPR (higher the better), and with 67% decreases in FPR95 (lower the better). On the other hand, ResNet50 achieves slightly better performance in the far-OOD scenario (right panel). Back to the comparison between ViT-B-16 and Swin-B, one sees comparable performance in AUROC and AUPR, with ViT model performs slightly better in FPR95. This might be due to shifted window introduced in Swin Transformer, which reduces its computational complexity but makes it slightly less capable in information extraction.

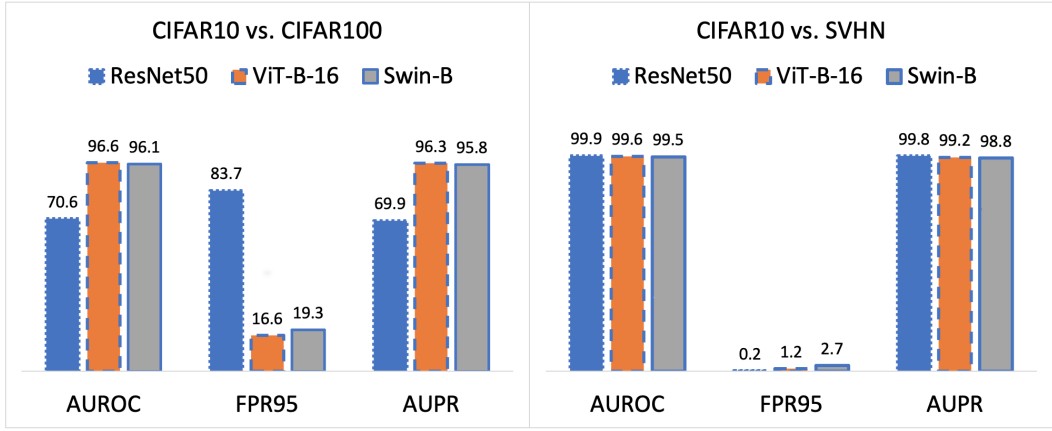

Figure 3: Performance comparison using ResNet50, ViT-B-16, and Swin-B.

### 4.4 DISTRIBUTIONS OF SEMANTIC SHIFTED AND COVARIATE SHIFTED DATASETS

We visualise and compare the ID and OOD distributions over the four datasets (CIFAR-10/100, SVHN and Texture) in feature space, in which CIFAR-10 is set as the ID set and the rest three are the OOD sets.

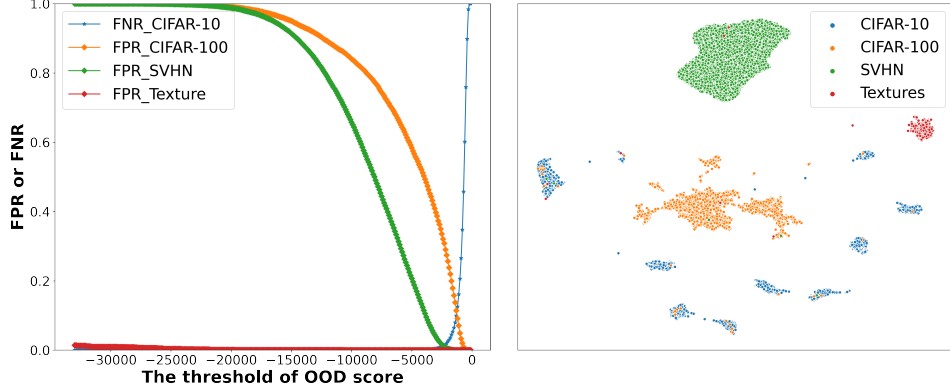

Figure 4: Visualisation of in-distribution and out-of-distribution. The left figure is the FNR of ID datasets and the FPR of OOD datasets under different threshold of OOD score, and the right one is visualization of embedding on feature space.

From the right panel of Fig. 4, one sees that both CIFAR-10 and CIFAR-100 shares some points and on the contrary SVHN points are more focal and locates further away from the CIFAR datasets. The left panel of Fig. 4 depicts the FNR and FPR of ID and OOD datasets separately in terms of the threshold of OOD score, which can illustrate the overlap level among different datasets. One observes that only CIFAR datasets are overlapped with each other obviously (about 16% of CIFAR-100 samples are mixed with CIFAR-10) while SVHN and Texture are slightly overlapped, even non-overlapped, with CIFAR-10. The main reason of these observations is that SVHN and Texture belong to domains further away from CIFAR-10. We also compare distributions of CIFAR-10/100 under different hyper-parameter settings, see Fig. 6 in Appendix A.1 for details.

### 4.5 THE ABLATION STUDY OF TEMPERATURE AND NUMBER OF CLUSTERS

In the absence of training data labels, we can divide the training dataset into $k$ clusters and evaluate whether the number of clusters $k$ affect the performance. The left panel of Fig. 5 shows that the model achieves good performance when the number of clusters is $k = 1$. The ID dataset (CIFAR-100) has 10 classes, thus we select 1 and 10 as number of clusters for temperature ablation. For

1 cluster case, the AUROC is maximised at 0.5 temperature and the lowest AUROC is at 0.07 temperature.

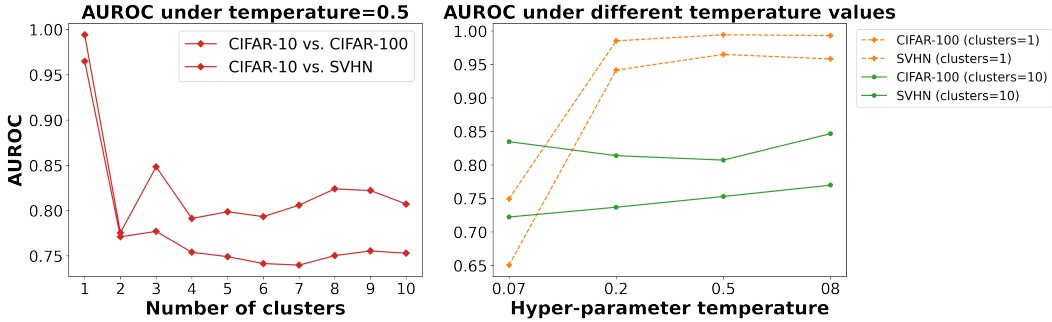

Figure 5: The left panel shows AUROC trends with 1 to 10 clusters and the right panel depicts the performance under different temperatures in single cluster scenario. CIFAR-10 is set as the ID set.

Table 3: AUROC with different temperature values in self-supervised training. The in-distribution dataset is CIFAR-10, the OOD datasets are CIFAR-100 and SVHN respectively

| Dataset | Number of clusters | Temperature (0.07) | (0.2) | (0.5) | (0.8) |
|---------|--------------------|--------------------|-------|-------|-------|
| CIFAR-100 | 1 | 65.05% | 93.14% | 96.48% | 95.81% |
| | 10 | 72.23% | 73.69% | 75.29% | 76.98% |
| SVHN | 1 | 74.94% | 98.51% | 99.41% | 99.29% |
| | 10 | 83.47% | 81.38% | 80.72% | 84.66% |

On the contrary, the AUROC on CIFAR-10 vs. SVHN is the minimum at 0.5 temperature when cluster number equals to 10, as shown in Table 3. The observed trend of accuracy is consistent to those in Fig. 5. In general, when we have one cluster and temperature is 0.5, the model gains the best performance.

## 5 CONCLUSIONS

In this paper, we proposed a new self-supervised OOD detector, the CVT model, which outperformed competing OOD detectors in benchmark tests over three different datasets (CIFAR-10/100 and SVHN). In the CVT model, we developed an ensemble module, which not only balanced the performance on far and near OOD datasets, but also enhanced overall OOD detection performance. The CVT model achieved a notable 80.34% AUROC accuracy in the challenging unlabeled far-OOD detection task. Although the results were remarkable, there exists open issues and promising directions for future work. Firstly, the adopted datasets are well established and recognised, but relatively constrained. Images in CIFAR-10, CIFAR-100, and SVHN normally contain only one object. Further evaluation of the proposed CVT model may requires more complicated and real datasets. Besides, with label information, a supervised OOD detector may learn more features than unsupervised learning. Hence, the supervised OOD detector is worth to be further explored and we believe the ensemble module in CVT model will also benefit supervised OOD detectors. Finally, the CVT model has a good potential to be integrated to other (non-OOD) self-supervised tasks, OOD detection enabled CVT model may save time and improve reliability of the other OOD self-supervised model.

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

# A APPENDIX

## A.1 ABLATION

### A.1.1 THE INFLUENCE OF TEMPERATURE AND NUMBER OF CLUSTERS

We visualize the features of ID and OOD datasets under different temperature parameters in Fig. 6. Here the ID dataset is CIFAR-10, and the OOD datasets are CIFAR-100 and SVHN respectively. When observing the divergence of CIFAR-10 under 0.07 and 0.5 temperatures, we find the clusters become more cohesive when the temperature is 0.5. From the pictures on CIFAR-10 vs. SVHN at temperature 0.07, the features from SVHN are farther from features from CIFAR-10 than features from CIFAR-100. Hence, temperature controls the distance among each sample in the feature space, while the contrastive loss makes the samples from the same class close and repel others in the feature space. A low temperature will spread features, and the features will be evenly distributed when it is small enough. Although a large temperature can make features from the same class more cohesive, it also shortens the distance between different classes.

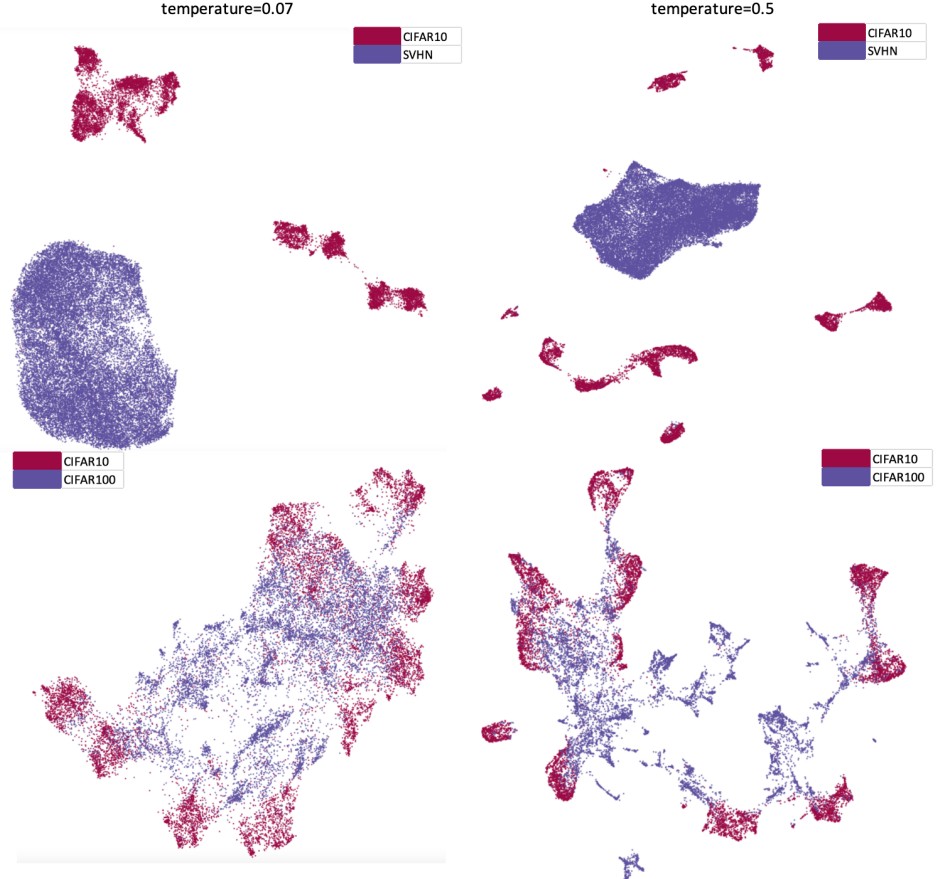

Figure 6: Visualization of features under different temperature parameters. The temperature of the two pictures on the left is 0.07, and the pictures on the right are 0.5.

### A.1.2 DIMENSION OF FEATURE EMBEDDINGS

To discover how many dimensions of feature embeddings are suitable for OOD score calculation, we experiment with different dimensions to compute the OOD score. Here, we use the features from predictor and encoder respectively to calculate metrics. As shown in Table 4, the AUROC begins with the lowest value at 128 dimensions on CIFAR-100 and SVHN OOD datasets when using features from predictor. It becomes stable when the dimension is larger than 256.

Table 4: AUROC with different dimension values in self-supervised training and the in-distribution dataset is CIFAR-10.

| OOD Dataset | Feature type | Dimensions (128) | (256) | (512) | (768) |
|---|---|---|---|---|---|
| CIFAR-100 | Predictor features | 95.05% | 96.48% | 96.61% | 96.56% |
| | Encoder features | 93.59% | 95.47% | 94.07% | 94.89% |
| SVHN | Predictor features | 99% | 99.41% | 99.42% | 99.62% |
| | Encoder features | 99.83% | 99.9% | 99.94% | 99.95% |

### A.2 THE IMPACT OF DATA AUGMENTATION ON OOD DETECTION

Considering that data augmentation with a few specific techniques is applied to ID dataset for generating positive pairs in contrastive learning, it is valuable to explore how the data augmentation impact OOD score, especially for the ID test set. In training process of our method, the Random-ResizedCrop in torchvision and the GaussianBlur in Pillow are utilised for data augmentation. The scale for cropping image is from 0.08 to 1, which is denoted by Crop(0.08, 1), whilst the radius of Gaussian kernel in GaussianBlur is from 0.1 to 2, which is denoted by GaussianBlur(0.1, 2). To explore whether the augmented samples from ID test set will still be recognised as ID ones, new parameters different from those in the training are set for these two techniques to generate augmented ID test set, and then the trained model is used to detect these new augmented samples. The results shown in Table 5 demonstrates that data augmentation has almost no impacts on the OOD score for ID samples.

Table 5: The impact of data augmentation on OOD detection. This toy experiment is conducted on two scenarios, where CIFAR-10 and CIFAR-100 are used as ID dataset respectively, and FPR95 is selected as the metric, where a higher FPR95 is, the less impact data augmentation has on OOD detection.

| ID dataset | Augmented ID test set | | |
|---|---|---|---|
| | Crop(0.03, 0.08) | GaussianBlur(2, 3) | Crop(0.03, 0.08) + GaussianBlur(2, 3) |
| CIFAR-10 | 99.98% | 99.76% | 99.98% |
| CIFAR-100 | 99.99% | 99.22% | 99.98% |

