# OpenReview forum: "Contrastive Vision Transformer for Self-supervised Out-of-distribution Detection"
_ICLR.cc/2023/Conference — Submitted to ICLR 2023_

### Official Review · Reviewer_RtUc · 2022-10-20

**Confidence:** 3
**Correctness:** 3
**Technical Novelty And Significance:** 2
**Empirical Novelty And Significance:** 2
**Recommendation:** 3

**Clarity, Quality, Novelty And Reproducibility:**

Clarity: good

Quality: fair

Novelty: poor

**Strength And Weaknesses:**

## Strength
### An ensemble module is developed to build representative ensemble features for achieving the balance between semantic and covariate OOD detection.

## Weaknesses
(1) The architectures and the modules in the proposed method are not novel. The four extra modules (except for the ensemble module) are mostly existing modules from previous methods, such as MOCO, MOCO v3, BYOL, et. al.
(2) The contributions of this paper are not significant. The major contribution is apply the contrastive learning architecture for OOD detection.

**Summary Of The Paper:**

This paper proposed a contrastive vision transformer for self-supervised OOD detection. An ensemble module is developed to build representative ensemble features for achieving the balance between semantic and covariate OOD detection. The experimental results show the effectiveness of the proposed method.

**Summary Of The Review:**

The novelty and the contributions of this paper are limited as indicated in the weaknesses. Thus, I tend to give a negative rating.

---

> ### Author Response · Authors · 2022-11-19
> **Response to the comments from Reviewer RtUc**
>
> We thank the reviewer for the feedback. We have clearly indicated in the second paragraph of page 2 that the proposed model does absorb the successful components/ideas from existing methods, and have also stated that the main contributions are (1) integrating ViT architecture under a contrastive learning framework we designed for specific OOD tasks, as well as (2) developing an ensemble feature for computing OOD score which not only balance but also enhance the performance of features on OOD detection. The proposed method achieves surprisingly good performance for self-supervised OOD detection in image classification tasks. We have also released the source code for readers to reproduce the interesting results.
>
> Although the proposed architecture is not a completely novel one, we believe this new design paradigm and its outstanding performance in the self-supervised OOD task is beneficial and worth to be reported to the community. The view on the strength of this paper is also shared by the other two Reviewers YUrT and nmrp in their reviews, both of them provided very useful comments for us to further improve the quality of this manuscript.
>
> Given the above, we kindly ask the reviewer to reconsider her/his position regarding the contributions of this paper and the given score.

---

### Official Review · Reviewer_nmrp · 2022-10-25

**Confidence:** 4
**Correctness:** 3
**Technical Novelty And Significance:** 3
**Empirical Novelty And Significance:** 3
**Recommendation:** 5

**Clarity, Quality, Novelty And Reproducibility:**

It's not clear to me how the other models in Table 1 were trained.  Were these models just trained as binary classifiers to predict ID vs. OOD?  How the metrics were defined for both the alternate and proposed approach in this table is unclear.

Fort et al. is shown to be significantly better than the proposed approach, but this approach is never discussed anywhere- attention needs to be given to this in the related works.

**Strength And Weaknesses:**

Strengths:
Identifying OOD data is an important topic.  The ability to identify OOD data using the strengths of SSL is a valuable endeavor.

Weaknesses:
I think it would be valuable to see how the choice/amount of augmentation impacts OOD score.  For example, if a sample is augmented with new/significant noise, will it be declared out of domain?

It's not clear to me from 3.3 and 4.1/2 how the other models in Table 1 were trained.  If they were just trained to predict ID vs. OOD, this doesn't seem like a realistic task since that will never be known in the real world.  Or were the models trained to carry out the normal prediction task and only if the overall class likelihood was near chance, was it declared OOD?   And in contrast is the proposed model scored using the new metric defined in (4)?

The authors reference VAEs as another common approach which has been successful in the past, but that is not explored here.  Why?

CIFAR is a small dataset (i.e. the images are small) so the queue size can be quite large on a "standard" machine.  If the batch size has to be reduced because "normal" sized images are used (512x512 or much larger), how will this impact performance and/or training/inference feasibility?

Why is the approach of Fort et al. not shown in Table 1?

The authors claim better than SOTA, but this isn't quite true.  The binary task isn't really "fair" at some level because ID vs. OOD is not known in advance.  The best comparison seems to be with Fort et al., but all related work and discussion around this is missing.  This omission feels intentional, but the authors need to discuss the strengths/weaknesses to each.

Comments:
The authors state a number of factors generating outliers.  In the context of self supervision, in particular, often "outliers" arise because no massive dataset exists for the task of interest.  The pretraining dataset may be a large public dataset like imagenet.  The learned representation is then transferred or fine-tuned to the domain/task of interest.  Can such an approach still be used in those scenarios.

What impact does augmentation procedure have on the ability to identify OOD samples?

How dependent is this approach on the specific constrastive learning approach selected?

The authors state three metrics are used in 3.3, but tables 1 and 2 only focus on AUROC.  What do the other metrics reveal?

Figure 4 seems confusing.  For as clean as the embedding clusters are on the right, there is still fairly significant overlap of the distributions on the left.

**Summary Of The Paper:**

The authors propose an attention-based contrastive learning model (CVT)  for OOD detection in image classification tasks based on self-supervised methods by using a ViT as a feature extractor.  The proposed model outperforms several SOTA methods based on CIFAR-10/100.

**Summary Of The Review:**

The authors propose a clear, relatively straightforward approach to identifying OOD data using a supervised framework.  Overall the writing is clear, but a few improvements could be made.  The impact is limited by the focus on small toy datasets (CIFAR 10/100, SVHN) as opposed to more real world datasets where such approaches would be needed.

 The results when compared against a binary ID/OOD task are much better, but this doesn't feel like a fair comparison since such information would never be known in advance; when comparing to the more standard task (Table 2) the results are still quite good, but not better than Fort et al.  Additional discussion needs to be provided to this key work.  The paper can still be accepted even if it does not beat Fort et al., but it can't be overlooked as it is currently since this is SOTA (not the approaches in Table 1, as the authors claim).

---

> ### Author Response · Authors · 2022-11-19
> **Response to comments from Reviewer nmrp - Part 2**
>
> 10. FPR95, AUROC and AUPR are three metrics for assessing the model's performance on OOD detection. AUROC is the most popular and comprehensive one (Sun et al., 2022; Tack et al, 2020). The highest AUROC is always corresponding to the lowest FPR95 and the highest AUPR whereas the lowest FPR95 is not corresponding to the highest AUROC. That's why we only focus on AUROC.
> 11. Sorry for the confusion. The 'fairly significant overlap' is due to an inappropriate display form. We replaced the left figure with a new one in Figure 4, where the FNR for the ID dataset (i.e., CIFAR-10) and the FPR for OOD datasets (i.e., CIFAR-100, SVHN, and Texture) is plotted over the threshold of OOD score. The new figure can illustrate the overlap level among different datasets more clearly. It shows that only about $16\%$ of CIFAR100 samples are mixed with CIFAR-10 samples. And the main reason for this is that some classes (e.g., 'bus', 'pickup truck', 'streetcar' and 'tractor') in CIFAR-100 belong to the super-class of 'automobile' in CIFAR-10.
>
> We hope our reply addressed the reviewer's concerns, if so we would appreciate it if the reviewer would consider increasing the rating.
>
> Xiao, Z., Yan, Q. and Amit, Y., 2020. Likelihood regret: An out-of-distribution detection score for variational auto-encoder. Advances in neural information processing systems, 33, pp.20685-20696.
> Yang, J., Zhou, K., Li, Y. and Liu, Z., 2021. Generalized out-of-distribution detection: A survey. arXiv preprint arXiv:2110.11334.
> Grill, J.B., Strub, F., Altché, F., Tallec, C., Richemond, P., Buchatskaya, E., Doersch, C., Avila Pires, B., Guo, Z., Gheshlaghi Azar, M. and Piot, B., 2020. Bootstrap your own latent-a new approach to self-supervised learning. Advances in neural information processing systems, 33, pp.21271-21284.
> He, K., Fan, H., Wu, Y., Xie, S. and Girshick, R., 2020. Momentum contrast for unsupervised visual representation learning. In Proceedings of the IEEE/CVF conference on computer vision and pattern recognition (pp. 9729-9738).
> Sun, Y., Ming, Y., Zhu, X. and Li, Y., 2022. Out-of-distribution Detection with Deep Nearest Neighbors. arXiv preprint arXiv:2204.06507.
> Tack, J., Mo, S., Jeong, J. and Shin, J., 2020. Csi: Novelty detection via contrastive learning on distributionally shifted instances. Advances in neural information processing systems, 33, pp.11839-11852.

---

> ### Author Response · Authors · 2022-11-19
> **Response to comments from Reviewer nmrp - Part 1**
>
> We appreciate the reviewer's insightful feedback. Below is our response to the reviewer's concerns.
> 1. Thank you for your valuable suggestion. We conducted an additional experiment (Appendix A.2) to explore how the data augmentation impact OOD score, especially for the ID test set. By using different parameters from those training ones for RandomResizedCrop and GaussianBlur in data augmentation, the new augmented ID test sets are generated and then the trained model is used to detect these new augmented samples. The final results (Table 5) demonstrate that data augmentation has almost no impact on the OOD score for ID samples.
> 2. Sorry for the confusion. All the models were trained to carry out the normal prediction task and the OOD samples are detected by some metrics (i.e., OOD score) applied to the learned representations. As for our model, yes, the metric defined in (4) was selected to compute the OOD score.
> 3. Citing VAE references is for completeness of the literature review as there exist some VAE-based methods for the OOD detection tasks. And please note that we didn't state that `the VAE methods have been successful in the past. In fact, Yang et al. (2021) reported that generative models (including VAE) often had worse performance than classification-based approaches (we mentioned this concept in the last paragraph of Section 2). Moreover, VAE-based OOD detection methods such as Likelihood Regret (Xiao et al. 2020) did not outperform competing methods for detection on several OOD datasets (e.g., SVHN) when CIFAR-10 is used as the ID dataset. Therefore, we chose the contrastive learning-based representative methods, some of which achieved state-of-the-art, for numerical comparison.
> 4. Thanks for the insightful question. If we understand your question correctly, the concern relies on: the increment of image size leads to a reduction of queue size, and a small queue size may degenerate model performance. In our proposed method, we resize all images to the size of 224*224 before model execution. This is because the size of the images from different datasets varies, but our CVT model employs a pre-trained Vision Transformer (with fixed input size 224*224) as a component in our model. Hence, the batch size and queue size in our experiments can be fixed to 64 and 4096 respectively. Under this setting, our model can achieve outstanding performance in the experiments for both small-sized (e.g., CIFAR-10 and CIFAR-100) and large-sized images (e.g., DTD). It demonstrates the robustness of the proposed CVT model, which adapts well to datasets with different image sizes when using a fixed batch size and queue size.
> 5. Our proposed method is not designed only for a binary classification between ID and OOD samples. We realised that this confusion may be raised from Table 1 and Table 2 in the original version of the manuscript, the two tables are now combined as Table 1 in the updated version. A quick answer is, the original Table 1 reported the main self-supervised learning OOD task results, mainly to demonstrate that our model is the best performed one in self-supervised learning OOD. In contrast, the original Table 2 was the extended evaluation task on supervised learning OOD. It aims to show that our model, which was trained only on the ID samples (exactly the same way as that in the self-supervised learning task, i.e., training without ID samples labels), could still outperform most of the supervised OOD detection methods except Fort et al. Nevertheless, Fort is a supervised method where the ID sample labels were available to their model. So the fact is that our proposed method is better than both the previous state-of-the-art self-supervised methods and most of the supervised methods. We have updated the results in the new Table 1 and hopefully this is clear now.
> 6. Replied in 5.
> 7. Actually, a fundamental component in our proposed method is the Vision Transformer which is pre-trained on a large public dataset. The training process of our model on some specific ID datasets can be treated as transferring or fine-tuning the learned representation to the domain/task of interest. Therefore, yes, our approach can still be used in such scenarios.
> 8. Replied in 1.
> 9. The specific contrastive learning approach with a fundamental structure of contrastive learning family was selected as the backbone of our proposed model, and it integrated an additional predictor from BYOL and a memory queue scheme from MoCo. Using this framework is aimed to extract robust features under a self-supervised learning regime. So the baseline to validate the performance of this framework is a strong feature extractor under a supervised learning regime. This has been explored in previous research (Grill et al, 2020; He et al, 2020).

---

### Official Review · Reviewer_YUrT · 2022-10-25

**Confidence:** 3
**Correctness:** 3
**Technical Novelty And Significance:** 2
**Empirical Novelty And Significance:** 2
**Recommendation:** 5

**Clarity, Quality, Novelty And Reproducibility:**

The paper can be reproduced. The originality of the work is low. Given that the paper repurpose existing methods with different encoders, the quality should come from rigorous evaluations, but the paper still needs to improve in this regard.

**Strength And Weaknesses:**

Strong Points:

Having a strong encoder is very relevant for OOD detection, thus having a contrastive method that works with ViT model is beneficial.

Weak Points:

Since the contribution of this paper is empirical, coming from replacing a convolutional model with a stronger, Transformer model, the evaluation should be more rigorous. At the moment only CIFAR10-100 and SVHN are used to form IN / OOD pairs. The setups from CSI and [A] should also be evaluated.

The claim that Encoder features and Predictor features are more appropriate in different settings is based on the experiments in Table 3. But these are not conclusive, as Encoder features are better in 3 of the 4 cases so it cannot be said that they are better for semantic or covariate shifts.

Multiple seeds should be used when computing the results and confidence intervals should be given.

[A] Sun et al. “Out-of-Distribution Detection with Deep Nearest Neighbors” ICML 2022


**Summary Of The Paper:**

The paper proposes to use a contrastive method that uses ViT models as encoders for out-of-distribution detection. Different from previous works, this paper uses contrastive methods with Vision Transformers encoders instead of convolutional ones. Different representations obtained by the model seem to be more appropriate to different settings (near vs far OOD) and they are combined. The resulting method obtains good results on OOD detection datasets.

**Summary Of The Review:**

The paper proposes to use a contrastive learning based on ViT models for OOD detection. It results in a good method, but it needs to be evaluated more rigorously.

---

> ### Author Response · Authors · 2022-11-19
> **Response to comments from Reviewer YUrT**
>
> We appreciate the reviewer's insightful feedback. Below is our response to the reviewer's concerns.
> 1. Our experiments were initially designed to benchmark model performance for scenarios in which other competing models reported relatively worse performance (e.g., other models performed poorly in the scenario when CIFAR-100 was used as an in-distribution dataset). We adopted the widely used CIFAR10 and CIFAR100, which were also the main parts of the evaluation datasets adopted in the papers you suggested.
>     But we agree with your comments, and add an extra Describable Texture Dataset (DTD) for model evaluation (as reported in the revised Table 1), which is intentionally selected from the popular OOD datasets (e.g., LSUN, DTD, Places-365). This is mainly because DTD contains large-size images, which is a typical representative and useful addition to the existing three evaluation datasets we have adopted. We re-run all models and update Table 1, the results are consistent with the other previously reported ID/OOD ones, and demonstrate the robustness of the proposed CVT model for large-sized OOD images.
> 2. We replaced the previous related conclusions with more objective descriptions on our results in Section 4.2, as follows.
>     "The first two columns in Table 3 show four test settings that are grouped into semantic shift scenarios (e.g., CIFAR-10/CIFAR-100 vs. SVHN) and covariate shift scenarios (e.g., CIFAR-10 vs. CIFAR-100). By using the features from either the Encoder or the Predictor to compute the OOD score for detecting OOD samples, the results display different trends under the above two scenarios. The race of the OOD detection under the covariate shift scenario between low-level features (i.e., the features from the encoder) and high-level features (i.e., the features from the predictor) ends in a draw. By contrast, low-level features dominate the OOD detection under the semantic shift scenario. Therefore, it is hard to decide which kind of features should be finally used to compute the OOD score."
> 3. We have added 5 random seeds for all experiments reported in the paper and also reported the numerical results using mean and standard deviation to reflect the fluctuation level of the results, as shown in Table 1.
>
> We hope our reply addressed the reviewer's concerns, if so we would appreciate it if the reviewer would consider increasing the rating.

---

### Decision · Program_Chairs · 2023-01-20

**Decision:**

Reject

**Justification For Why Not Higher Score:**

Limited Novelty
Limited Empirical evaluation

**Justification For Why Not Lower Score:**

N/A

**Metareview: Summary, Strengths And Weaknesses:**

This paper proposes contrastive vision transformer for self-supervised OOD detection. While the paper has some interesting contributions, the main concerns from the reviewers were the limited novelty of the proposed approach and the limited empirical evaluations. Overall, the current version falls below the acceptance threshold. I encourage the authors to revise and resubmit to a different venue.